# Prevalence of hypertension and factors associated with the utilization of primary health care services for hypertension among hypertensive population aged 40 years and above in Pyin Oo Lwin Township, Myanmar

**May Sabai Soe**[1¤]*, **Su Su Hlaing**[2], **Aye Sandar Mon**[3], **Kyaw Thu Lynn**[1]

1 Pyin Oo Lwin District Public Health Department, Department of Public Health, Ministry of Health, Pyin Oo Lwin, Mandalay, Myanmar, 2 Department of Epidemiology, University of Public Health, Ministry of Health, Yangon, Myanmar, 3 Department of Biostatistics and Medical Demography, University of Public Health, Ministry of Health, Yangon, Myanmar

¤ Current address: Department of Public Health, Ministry of Health, Nay Pyi Taw, Myanmar
* maysabaisoe.mm@gmail.com

**Data Availability Statement:** All data are in the manuscript and Supporting information files.

## Abstract

### Background

Utilization of hypertension services at primary health care levels has not been assessed at township level, since launching of PEN interventions in Myanmar. This study aimed to determine the factors associating with the utilization of primary health care services for hypertension among 40 years and above hypertensive population.

### Methods

Community-based cross-sectional study was done in Pyin Oo Lwin Township, 2023. Multi stage sampling was conducted to recruit 40 years and above participants; response rate was 85%. Joint National Committee (JNC7) classification was used to define hypertension. Among hypertensive participants, descriptive analysis, Chi squared test and multiple logistic models were conducted, with a significance level of 0.05.

### Results

Out of 1001 screening participants, prevalence of hypertension was 38.6% (386). Among 386 participants, 51.8% (200) utilized primary health care services provided by public health facilities. Rural residents (AOR = 2.79, CI = 1.68, 4.67), known hypertension (AOR = 4.36, CI = 2.39, 8.23), good perception on hypertension (AOR = 0.30, CI = 0.14, 0.62), perceived cost of travel as necessary (AOR = 0.57, CI = 0.35, 0.92) and awareness of available services (AOR = 4.11, CI = 2.55, 6.71) were associated with the utilization of primary health care services for hypertension.

**Funding:** The authors received IR grant from Department of Medical Research, Ministry of Health, Myanmar. However, the funders had no role in study design, data collection and analysis, decision to publish, or preparation of the manuscript.

**Competing interests:** The author(s) declared no potential conflicts of interest with respect to the research, authorship, and/or publication of this article.

## Conclusion

This study provided context-specific scientific evidence to tackle existing problems of low utilization of PHC services for hypertension. Strengthening health care infrastructure for quality hypertension care at primary health care level was also recommended.

## Introduction

Hypertension, one of the non-communicable diseases (NCDs), is a major risk factors for cardiovascular diseases (CVD): ischemic heart disease, heart failure and stroke [1]. Myanmar has experienced increasing disease burden of hypertension and its related mortality [2–4].

As primary health care (PHC) is patient-centered, community-based and sustainable, implementation of interventions for chronic diseases like hypertension is feasible and applicable at primary care level [5]. Most patients with hypertension can be managed appropriately with simplified standard protocols at the primary health care facilities [6]. Since PHC serves as the first entry point to the health system, utilization of the PHC services results great outcomes, efficiency, accessibility and cost-effectiveness [5,7,8].

In Myanmar, public hospitals are mainly equipped for inpatients, overcrowded and understaffed enough to provide primary care for high volume of out-patients [9,10]. Meanwhile, PHC facilities serve as first service points, make risk-based approach to filter patients and reduce overflow to already loaded public hospitals [11]. As for private facilities, cost-effectiveness and accessibility to health services are jeopardized, when patients are charged even for primary health care services with direct out-of-pocket (OOP) payment [10,12].

To tackle various problems related to hypertension, Myanmar has adopted the package of essential non-communicable disease intervention in primary health care project (PEN project) to decentralize NCD care in primary health care settings [13]. PEN intervention was implemented through basic health staff (BHS) of public health facilities for major NCDs, including hypertension. They provide population-wide and individual-based interventions for hypertension which include screening for diagnosis and treating with affordable technologies and medications; and referring patients with complications to nearest secondary hospitals [14]. Thereafter, blood pressure measurement and essential medicines for hypertension were generally available in primary health care facilities of the public sector [15].

However, the prevalence of hypertension and uncontrolled hypertension have not met yet with the global target [1]. Prevention and control services for hypertension are still underutilized; despite Ministry of health of Myanmar emphasize on service availability and readiness for hypertension through implementation of PEN guidelines [2,16].

Since launching PEN project, utilization of hypertension services has not been assessed at township levels. Addressing demand-side determinants of the utilization would help formulate policies to improve the uptake of health services, particularly in underutilized areas. Although globally generated protocols were provided in formulating policy and strategies, PHC-oriented research were needed to synthesize country-specific evidence-based decisions to ensure delivery of quality and safe PHC services.

To generate such evidence, we conducted this study regarding to the utilization of PHC services concentrating on hypertension, and in particular those targeted at the population level. Therefore, this study aimed to assess the prevalence of hypertension, find out proportion of public primary health care services utilization for hypertension and determine the factors associating with the utilization of PHC services among the hypertensive population aged 40 years and above in Pyin Oo Lwin Township, Myanmar.

## Materials and methods

### Study design

A community-based cross-sectional study was carried out in Pyin Oo Lwin Township, Myanmar during September—November, 2023. Pyin Oo Lwin lies in the northeast of Mandalay Region in the central plain area of Myanmar. Approximately 220,000 population is covered by: one township public health department under which one maternal and child health care center (MCH), seven rural health centers (RHCs) and twenty-nine sub-rural health centers (Sub-RHCs); two station hospitals; and one district hospital.

Basic health staff are positioned in health facilities under township public health department. With PEN expansion, almost all of the basic health staff of Pyin Oo Lwin Township Public Health Department have got cascade training to provide NCD control and care to the population. Trainings about the interventions of PEN protocols are limited to Ministry of Health [17]. So, essential NCDs care services embodied at primary health care level (PEN protocols) are provided by public PHC facilities.

### Sample size

The sample size was estimated using 95% confidence interval, an acceptable error (alpha) of 5%, margin of error 4% and the prevalence of utilization of primary health care facilities for hypertension was 88% [11].

$$n = \frac{z^2 pq}{d^2} = \frac{(1.96)^2 (0.88)(0.12)}{(0.04)^2} = 254 \sim 290 (\text{including } 15\% \text{ non-response rate})$$

So, the minimum required sample size to interview was estimated as 290.

According to WHO base line estimates used in national strategic plan for NCDs (2017–2021) Myanmar, hypertension prevalence was 28.9% [16]. To find out the number of at least 290 hypertensive population, approximately 1001 participants needed to be screened.

$$n = \left(\frac{290}{28.9}\right) \times 100 \approx 1001$$

After screening the study participants, 386 participants were found to be hypertensive and all were taken for interviews.

### Study population and sampling procedure

A multistage sampling was used to select a representative sample of 40 years and above populations from the community. According to the urban-rural ratio (3:7) of the population of the township [18], 3 out of 10 wards were randomly selected as urban. There were 7 RHCs which covered the population from rural areas in the township. One village under the catchment area of each RHC was randomly selected as rural. Secondly, households which had 40 years and above population were listed from selected 3 wards and 7 villages. Thereafter, 100 households each were randomly selected from the household lists, resulting 300 households from urban and 700 households from rural.

While visiting the community, households that were refused to participate were excluded. Absent households were re-visited a second time on the same day to ensure maximum participation but refused households were not visited again. While experiencing vacant households, they were replaced with the most nearby households on the right and not previously selected, which had at least 40 years and above household member. A total of 54 households were replaced. From all consented households, all 40 years and above household members were

recruited for screening of high blood pressure after excluding those who were seriously ill, pregnant, unable to communicate (unable to listen and talk), unable to consent and unwilling to participate. Total 1183 participants from 810 households responded and 1001 respondents participated in screening.

During screening, after taking a rest for at least 5 minutes blood pressure (BP) was measured twice which was 10 minutes apart, in a sitting position, using WHO certified Omron (HEM-7120) digital BP cuffs and following American Heart Association (AHA) BP measuring guidelines [19]. Hypertension was defined as SBP $\geq$ 140 mmHg or DBP $\geq$ 90 mmHg or both at the average of two measurements, or with the self-reported adherence of antihypertensive medication by using Joint National Committee (JNC7) classifications [20]. Among the screening participants (n = 1001), 386 participants were found to be hypertensive. All those hypertensive participants were recruited for face-to-face interviews. At last, 362 participants were included in the final analysis after exclusion of 24 participants with missing information.

## Data collection methods and tools

Data collection time was from September 25[th], 2023 to November 30[th], 2023. Data was collected by using digital BP cuffs and a structured and pre-tested interviewer administered questionnaire developed in Kobo platform. Before the start of data collection, ten auxiliary midwives who were not working in the study area were provided 2-day trainings for data collection, using Kobo collect mobile application and measuring blood pressure systematically following AHA BP measuring guidelines [19].

Questionnaires consisted of two parts: screening questions and interview questions. Screening questions included age, sex, residences, history of hypertension approved by any medical practitioners and adherence to medications at least within 30 days. The second part of the questionnaires were only used for hypertensive participants and included background characteristics, individual health-related, health facility-related and utilization-related questions, which had been developed based on literature and validated questionnaires from other studies [11,21,22].

Questionnaires also included 20 questions about the knowledge on control and complications of hypertension, 10 questions for perception on hypertension and 8 questions for perception on PHC facilities including quality of care and health providers, attitude of health providers, long waiting time, availability of drugs, being treated with or without respect, being in good terms or having relationship with providers. In the knowledge section, every unprompted correct answer was granted 2 points and each prompted correct response was 1 point, prompted incorrect response, 0 point and "Don't know", 0 point. For both perception sections, 4-point Likert scale was adopted in which: 1 "Strongly disagree", 2 "Disagree", 3 "Agree" and 4 "Strongly agree" for items except for "Increasing salt and sugar intake is beneficial for health" and "Taking treatment regularly can make hypertension a chronic disease" where the scale was reversed. Computed scores were graded into low ($<50$[th] percentile), average (50[th]– 75[th] percentile) and high levels (above 75[th] percentile) of knowledge. Both perceptions are classified as poor ($<50$[th] percentile), average (50[th]– 75[th] percentile) and good (above 75[th] percentile).

Pre-testing of the questionnaires was carried out with individuals who were not part of the sample in order to validate the understanding and clarity of the items and necessary modifications were performed. During conducting data collection, data set was checked daily to reduce the occurrence of missing and errors.

## Outcome measurements

Utilization of primary health care services for hypertension provided at public health facilities is regarded as being screened or treated or taking medication or going follow-up at PHC health facilities or outreach or mobile clinics or by basic health staff within 6 months for hypertension and registered in department patients' registry.

PHC facilities are public health facilities which are mainly responsible for preventive services and public health activities under Department of Public Health, Ministry of Health, Myanmar, naming urban health center (UHC), maternal and child health center (MCH), rural health center (RHC) and sub-rural health center (Sub-RHCs).

Basic health staff (BHS) are public health staff–public health supervisors 1 and 2, midwives, lady health visitors (LHV), health assistants (HA), township health nurses (THN), township health assistants (THA), and township medical officers (TMO) who are providing all primary health care services at the township level under Ministry of Health.

People who live within the catchment areas of a public health facility, go to those facilities for health services. According to the national guidelines for BHS, patients–who aged 40 years and above come to the facilities or mobile clinics or outreach for any kinds of illness–get BP measured by BHS for screening and get registered into NCDs screening books.

Newly or previously diagnosed hypertensive cases of any ages: who get treatment: visit follow-ups: or get referred to higher level facilities, are summarized daily and entered into lists of hypertension cases–called NCDs registered books.

The outcomes, utilization of PHC services for hypertension, were measured by coinciding patients' self-report with registered books at the time of collection.

## Statistical analysis

The collected data were extracted from kobo collect platform in excel form and exported into statistical software R (4.3.1 version), where data cleaning, editing and coding was done. Exploratory data analysis was performed to check for missing values and influential outliers. After that, all variables were grouped into respective categories. Categorical variables were expressed as frequency (percentage). Chi-squared analyses between the independent variables and the outcome variable were done, for more information, see S1 Table.

Factors having $p<0.2$ in bivariate analyses were exported to a multiple logistic regression model as full model. Final model was obtained by applying backward elimination method. Variables at the significance level of 0.05 were retained in the final model, for more information, see S2 Table. Assumptions for multicollinearity were checked. There was no collinearity with VIF < 5 between the variables in all models. Hosmer-Lemeshow goodness of fit test was passed and Akaike Information Criterion (AIC) method were applied in all models to select the optimum model.

## Ethical considerations

This study was conducted through the permission of Institutional Review Board of University of Public Health, Yangon (UPH-IRB (2023/MPH/10)). The data collection was done only after thorough explanation of purpose of the study to participants and obtaining written informed consent. All the resultant hypertensive participants were referred to the nearest primary health care facilities. The confidentiality of the study participants was maintained throughout the data collection. After data collection, the participants were deidentified by removing unnecessary direct identifiers and encoding addresses before analysis.

**Table 1. Age-specific, gender-specific and residence-specific prevalence of hypertension among the study participants (n = 1001).**

| Variables | Hypertension (n = 386) | No Hypertension (n = 615) | p-value |
|---|---|---|---|
| **Age (years)** | | | <0.001 |
| 40–49 | 76 (21.3%) | 280 (78.7%) | |
| 50–59 | 110 (40.1%) | 164 (59.9%) | |
| >60 | 200 (53.9%) | 171 (46.1%) | |
| **Gender** | | | 0.1260 |
| Male | 125 (42.4%) | 170 (57.6%) | |
| Female | 261 (36.9%) | 445 (63.1%) | |
| **Place of Residence** | | | 0.2875 |
| Urban | 128 (41.2%) | 183 (58.8%) | |
| Rural | 258 (37.4%) | 432 (62.6%) | |

## Results

### Prevalence of hypertension

Among 1,001 screening participants, 386 (38.6%) were found to have hypertension. Prevalence of hypertension increased with age. Prevalence of 40–49 years age group was 21.3%; 50–59 years age group, 40.1% and ≥60 years age group, 53.9%. Regarding the gender, 42.4% of male and 36.9% of female had hypertension. Based on urbanity, 41.2% of urban residents and 37.4% of rural residents were hypertensive (Table 1).

### Background characteristics

Table 2 showed background characteristics of the study participants. Age of the study participants (n = 386) ranged from 40 years to 91 years. Mean age was 59.8 ± 11.1 years. More than

**Table 2. Background characteristics of the study participants (n = 386).**

| Variables | | Frequency (%) |
|---|---|---|
| **Age group (years)** | 40–49 | 76 (19.7%) |
| | 50–59 | 110 (28.5%) |
| | ≥ 60 | 200 (51.8%) |
| **Gender** | Male | 125 (32.3%) |
| | Female | 261 (67.7%) |
| **Residence** | Urban | 128 (33.2%) |
| | Rural | 258 (66.8%) |
| **Marriage** | Currently Married | 251(65.1%) |
| | Not Currently Married | 135 (34.9%) |
| **Education** | < High School | 352 (91.1%) |
| | ≥ High school | 34 (8.9%) |
| **Occupation** | Employed | 218 (56.5%) |
| | Unemployed | 168 (43.5%) |
| **Monthly Family Income (n = 362)** | ≤150,000 MMK | 103 (26.7%) |
| | >150,000 MMK | 259 (67.1%) |
| **Ethnicity** | Bamar | 312 (80.8%) |
| | others | 74 (19.2%) |
| **Religion** | Buddhist | 374 (96.8%) |
| | Others | 12 (3.2%) |

MMK = Myanmar Kyat.

two-thirds of the participants were female, from rural and being currently married. Most of the participants were Burmese, Buddhist, and below high school level in education. Monthly family income was ranging from 10,000 kyats to 2,800,000 kyats. Over two-thirds were having over 150,000 MMK monthly.

## Utilization of PHC services for hypertension

Out of 386, 200 participants with hypertension [51.8% (95% CI: 47%-57%)] utilized primary health care services at least once within 6 months. In bivariate analysis, utilization of the services was associated with 8 variables: place of residence; education; social or financial support; known status of hypertension; perception on hypertension; presence of public health facilities in their villages or wards; perceived cost of travel to their nearest public health facilities; and awareness of available hypertension services provided at public health facilities. Knowledge about control of hypertension and perception on public health facilities had no association with the outcomes. All responded that they were convenient with the clinic hours of the public health facilities.

After backward elimination (Fig 1), factors associated with the utilization of primary health care services for hypertension were rural residents ((AOR = 2.79, 95% CI = 1.68, 4.67) compared with urban residents, known hypertension (AOR = 4.36, 95% CI = 2.39, 8.23) compared with unknown hypertension, good perception on control and complications of hypertension (AOR = 0.30, 95% CI = 0.14, 0.62) compared with poor perception, necessary perceived cost of travel (AOR = 0.57, 95% CI = 0.35, 0.92) compared with those who perceived cost of travel as not necessary and having awareness of available services (AOR = 4.11, 95% CI = 2.55, 6.71) in comparing with those who were not aware of the services.

## Discussion

This study mainly focused on 40 years and above population with hypertension to explore the factors affecting utilization of primary health care services for hypertension among the targeted population.

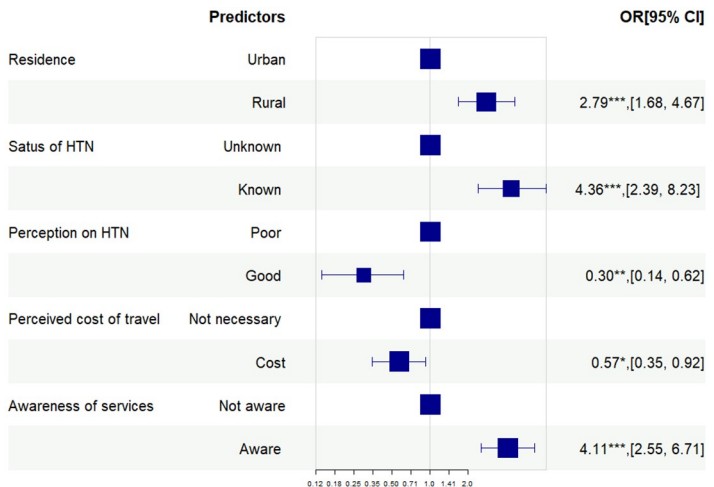

**Fig 1. Factors associated with the utilization of PHC services by multiple logistic regression (final model).** Pvalue —*** <0.001, ** <0.01, * <0.05.

## Prevalence of hypertension

The overall prevalence of hypertension (38.6%) was higher than the country's prevalence rates of 30.1% and 26.4% reported by 2009 and 2014 Nationwide STEPs surveys, respectively [23,24]. However, the age composition differences attributed to the relatively higher prevalence of the current study. The participants in the current study were aged ≥40 years while those of 2009 survey ranged 15–64 years, and of 2014 survey ranged 26–64 years. Moreover, the current study prevalence was not similar to the age-standardized prevalence rate of Myanmar (38% among 30–79 years) [1].

Prevalence comparison was made with other countries from South-East Asia Region (SEAR) with similar socio-economic backgrounds after age-standardization. The prevalence of hypertension in this study was higher than that of those studies done in Cambodia [25] and Bangladesh [26]; similar to Nepal [27] and Malaysia [28]; but lower than Vietnam [29], Indonesia [30] and Sri Lanka [31]. However, the overall prevalence might be underestimated because only one-thirds of the screening participants were male population and proportion of behavioral risk factors for NCDs–alcohol and tobacco (both smoke and smokeless)–were higher in male population of Myanmar [24]. Low participation of male in this study–which can also be seen in other community-based, cross-sectional studies [30,32]–reflected health seeking behaviours of male population. They were more likely to be occupied with busy schedules of work during day time and unmotivated to seek health care while they were not having any symptoms. Thus, NCDs care for economically active population at primary health care level should be prioritized to strengthen early detection and timely treatment.

After doing the age-specific analysis of the prevalence increased with age as the age itself was a strong risk factors for hypertension. Higher prevalence among older age group was consistent with the previous nationwide STEPs surveys [23,24].

Like other studies [26,33], urban had higher hypertension prevalence. It was concomitant with the findings detected in previous STEPs survey; greater proportion of metabolic risk factors–obesity, diabetes, hypertriglyceridemia and hypercholesterolemia–were found among urban residents in comparing with rural residents [34].

## Utilization of PHC services

A cross-sectional study was conducted among 386 hypertensive participants. Half of the participants (51.8%) utilized PHC services for hypertension offered by public PHC facilities. This utilization rate was somewhat higher than a study from Ghana (41.0%) [35]; but lower than that from China (88.4%) [11]. This increase in China was the benefits from the popularized China rural basic medical insurance system [36]. Thailand has also increased utilization of PHC services after implementing UCS (Universal health care coverage scheme) [37]. So, Myanmar needs to consider the proven interventions to increase the utilization of PHC services for hypertension in the future.

Other than that, there may be several reasons for low utilization of PHC services for hypertension, but possible reasons in this context were health seeking at higher level quality of care [11,38], shortage of human resources [13], and intermittent supply of essential medicine and equipment [13,39], which distract the goals of providing screening, diagnosis and treatment services for hypertension at the primary care level. Thus, the country also needs to increase its investment on public PHC system for essential health care infrastructure to optimize patient outcomes.

## Factors associated with the utilization of PHC services

As shown in Fig 1, main factors influencing the utilization of PHC services for hypertension were place of residence, known status of hypertension, perception on hypertension, perceived travel cost and awareness of hypertension services available at PHC facilities.

Urban residents were less likely to utilize the PHC services, which agreed with a China study [40]. There may be several reasons for lower utilization among urban residents. First, urban population have more access to health care services, since various levels of health care facilities were concentrated in urban areas [41]. Second, mandatory health insurance system, and tiered diagnosis and treatment system have not yet been well-established in Myanmar [9]. Third, urban residents choose health facilities based on their conveniences; which have geographical accessibility, affordability and sensitivity to their needs that meet to their satisfaction [42]. For rural, because of limited number of private facilities and less access to higher level health care facilities in comparing with urban areas, rural residents mainly rely on public PHC facilities.

For participants with unknown hypertension, a low level of overall education among the study participants may lead to their low level of knowledge, which contributed to lack of awareness about hypertension and low utilization of health services [43].

In case of utilization of PHC services, education also had negative association with the utilization of public health facilities in LMICs [35,40]. Besides, participants with good perception were more likely to be educated [44]. They may prefer higher level health facilities because they perceived them as offering good quality health care [35,36]. On the contrary, participants with poor perception—more likely to be less educated and poor -may enjoy primary health care services provided at public health facilities which required no or less OOP payments [45]. This indicates the urgent need to prioritize for improvement of essential health care infrastructure that ensures quality hypertension care in public primary health care system.

Although travel time to the health facilities did not have significant association with the utilization, participants who perceived cost was needed to commute to health facilities were less likely to utilize those services. This finding concurred with findings from other study [46].

Previous study from Malaysia also reported lack of awareness of health services as barrier in utilizing health care in their settings [46]. If they were aware of services provided at PHC facilities, and benefits of that services, that would impact on the utilization of those health facilities [47]. That can be seen in Bangladesh service delivery model, where community awareness of services led to increased use of primary health care services [48]. However, "no nearby health facility" may still play a main role for utilizing those services even if they were aware of those services.

Reportedly, all hypertensive participants were convenient with opening hours of public health facilities; it contradicts with many other studies [49,50]. Possible reason was that patients can see BHS even in after-hours and weekends when in need; but they may use them for other services like ante-natal care, post-natal care and other common health problems. Further study to explore community health behaviors related to hypertension care, as well as, social factors are needed to conduct.

This community-based study highlighted the need for promoting public knowledge about hypertension; scaling-up dedicated hypertension care along with quality assurance of public health facilities to meet the expected needs of the population; extending service delivery points to improve equitable access and financial protection, especially for urban slum and rural remote area; and awareness raising for available PHC services. Moreover, prioritization of hypertension care among underutilized population—mostly economically active population—was needed to reduce high burden of diseases in the future.

This study also had few limitations. This study was subject to biases related to selection and information retrieval, which may have effects on prevalence estimates and association measures. First, lower participation of male may underestimate the overall prevalences, though the study had good response rate (85%).

Second, this study did not include private primary health care facilities, e.g., GP clinics because this study focused on PEN implemented primary health care settings and this intervention was limited to public health facilities. This may underestimate the utilization rate of PHC services for hypertension. Future studies should compare utilization of public primary and private primary facilities to better understand utilization patterns at the primary healthcare level.

Third, social desirability biases may also be generated because of face-to-face interview, even though all interviewers were well trained. Thus, perceived health status was mostly rated as good while half of them were having at least one comorbidity. Furthermore, perception on health facilities showed no significant relationship with the utilization of PHC services given by those facilities although many studies reported them as reasons for not utilizing health care in their settings [46,50]. Lastly, the findings may not be generalizable to the whole country, but may reflect the challenges of 40 years and above hypertensive population residing in townships with same socio-demographic characteristics in accessing health care.

## Conclusions

This study found out high prevalence rate of hypertension and low utilization rate of PHC services for hypertension among 40 years and above population which was not efficient for both patients and the health system. Findings from this study provided context-specific scientific evidences to tackle existing problems of low utilization of the PHC services for hypertension. Undoubtedly, service availability and readiness regarding prevention and control of hypertension services must be adequate beforehand.

## Supporting information

**S1 File. List of abbreviations.**
(DOCX)

**S1 Table. Bivariate analysis between independent variables and the utilization of PHC services among the hypertensive study participants (n = 386).**
(DOCX)

**S2 Table. Factors associated with the utilization of PHC services by multiple logistic regression models.**
(DOCX)

## Acknowledgments

The authors would like to thank to all study participants, data collectors and University of Public Health, Yangon for giving ethical clearance.

## Author Contributions

**Conceptualization:** May Sabai Soe, Su Su Hlaing, Aye Sandar Mon, Kyaw Thu Lynn.

**Data curation:** May Sabai Soe.

**Formal analysis:** May Sabai Soe, Su Su Hlaing, Aye Sandar Mon.

**Funding acquisition:** May Sabai Soe.

**Investigation:** May Sabai Soe, Su Su Hlaing.

**Methodology:** May Sabai Soe, Su Su Hlaing, Aye Sandar Mon, Kyaw Thu Lynn.

**Project administration:** May Sabai Soe, Su Su Hlaing.

**Resources:** May Sabai Soe, Kyaw Thu Lynn.

**Software:** May Sabai Soe, Su Su Hlaing.

**Supervision:** May Sabai Soe, Su Su Hlaing, Aye Sandar Mon.

**Validation:** May Sabai Soe, Su Su Hlaing.

**Visualization:** May Sabai Soe, Su Su Hlaing, Aye Sandar Mon.

**Writing – original draft:** May Sabai Soe, Su Su Hlaing, Kyaw Thu Lynn.

**Writing – review & editing:** May Sabai Soe, Aye Sandar Mon.

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
