## [Decision Letter · Decision Letter 0]

30 Jul 2024

PONE-D-24-18219Prevalence of hypertension and factors associated with the utilization of primary health care services for hypertension among hypertensive population aged 40 years and above in Pyin Oo Lwin Township, MyanmarPLOS ONE

Dear Dr. Soe,

Thank you for submitting your manuscript to PLOS ONE. After careful consideration, we feel that it has merit but does not fully meet PLOS ONE’s publication criteria as it currently stands. Therefore, we invite you to submit a revised version of the manuscript that addresses the points raised during the review process.

We look forward to receiving your revised manuscript.

Kind regards,

Kyaw Lwin Show, MPH, PhD

Academic Editor

PLOS ONE

Journal Requirements:

"The author(s) declared no potential conflicts of interest with respect to the research, authorship, and/or publication of this article."

Additional Editor Comments:

Congratulations to the authors for their hard work. Please find my suggestions for improvement below.

- The introduction needs to be restructured and improved. Furthermore, connections between paragraphs are not smooth. Information regarding the important role of primary care services in controlling hypertension is missing.

- You mentioned 1001 participants, but from how many households? Was there any household replacement?

- Can you differentiate whether the utilization is at facilities or through outreach according to the data? This information is important for public health interventions. For e.g. many of the participants utilized through outreach, then outreach interventions should be scaled up.

- Did you cover general practitioners (GPs)? Since GPs are important primary care providers in the community, if not, please consider changing the focus to 'utilization of public primary care services' and acknowledge the limitation of not covering GPs.

- Please response to reviewers’ comments

Reviewers' comments:

Reviewer's Responses to Questions

**Comments to the Author**

1. Is the manuscript technically sound, and do the data support the conclusions?

Reviewer #1: Yes

Reviewer #2: Partly

Reviewer #3: Yes

2. Has the statistical analysis been performed appropriately and rigorously? 

Reviewer #1: Yes

Reviewer #2: Yes

Reviewer #3: Yes

3. Have the authors made all data underlying the findings in their manuscript fully available?

Reviewer #1: Yes

Reviewer #2: Yes

Reviewer #3: Yes

4. Is the manuscript presented in an intelligible fashion and written in standard English?

Reviewer #1: Yes

Reviewer #2: Yes

Reviewer #3: Yes

5. Review Comments to the Author

Reviewer #1: 1. The abstract's methods section briefly describes the study design but lacks important details like how participants were selected and the response rate.

2. The rationale for determining the sample size is well-explained, considering factors like confidence interval and margin of error. However, there's a discrepancy between the reported prevalence rates (88% vs. 28.9% from national estimates). To ensure accuracy, it's important to clarify how these differences were accounted for in the sample size calculation.

3. The sampling procedure should address any potential biases introduced during participant selection.

4. The definition of primary health care service utilization for hypertension is clearly defined, covering different types of health facilities and services. To strengthen this section, it would be beneficial to provide more details or validation methods on how utilization was measured in the health system

5. It would be helpful to briefly mention any steps taken to protect participant confidentiality and ensure data security during and after data collection.

6. It would be good to clarify whether the study used age-standardization when comparing hypertension prevalence rates with other studies.

7. It would be better to expand on how these findings could inform the improvements of broader health policy and primary healthcare system in Myanmar.

8. Comparing findings with studies from other countries adds depth to the discussion. However, it's important to clearly explain how these comparisons were made, considering differences in healthcare systems and socio-economic factors.

Reviewer #2: Thank you very much for your effort in adding new information and evidence to the field of non-communicable diseases (NCDs), especially in resource-limited developing countries like Myanmar.

In the attachment document, I have included suggestions for revision in relevant areas.

In Financial Disclosure, it is mentioned that “The author(s) received no specific funding for this work”. However, the authors acknowledged the IR grant committee of Department of Medical Research, Ministry of Health. Could you please clarify whether this study was funded by the grant from IR grant committee of Department of Medical Research, Ministry of Health? If not, could you explain how the expenses for this study were covered.

I would like to suggest a through proofread. Some sentences are lengthy and could be rephrased for better clarity and readability.

The abstract should serve as a standalone summary, including all essential details for comprehension without referring to the full manuscript. In this abstract, methods need to be elaborated further, and results and conclusion should be aligned.

Line 29: Please specify the year in which stroke was the first leading cause of death in Myanmar

Line 42: Please correct the grammar for “PHC-oriented research were needed”

In Introduction section, it would be better to explain the reasons for choosing Pyin Oo Lwin Township and the population aged 40 years and above. Additionally, please clarify whether PHC services users need to pay for consultation fees or medicine.

Line 51-59: it is better to provide descriptions of other PHC services providers in study site such as GP clinics, or outpatient departments which are not under the township public health department and should explain the reasons why these are not included as PHC services providers in this study.

Line 71-79: It would be helpful to provide description of the location of the 7 villages and 3 wards.

Line 98: “Data was collected” should be “Data were collected”

Line 104: “two-parts” should be “two parts”

Line 161: Please clarify “not aware of their statuses”. I assume “not aware of their hypertension”.

Line 168: “Table (1) showed” should be “Table (1) shows”

Line 173: Could you please clarify the rationale for using 150,000 MMK as the cutoff point?

Line 181: “Shown in figure (2)” should be “As shown in figure (2)”.

Line 185-186: Please clarify high perception and low perception.

In Results section, it would be better to include relevant findings from bivariate analysis. Additionally, important negative findings-factors not associated with utilization of PHC services-should also be described and discussed in the main manuscript.

Line 195: “Age composition” should be “The age composition”

Line 199: “30-79Years” should be “30-79 years”

Line 207: Can you please spell out “SEAR”

Line 220: Please spell out LHV

Line 226: I would like to suggest using “However” instead of “But” at the beginning of the sentence.

Line 248-249: Could you please elaborate more on “Consequently, urban residents choose health facilities based on their conveniences.”

Line 270: I would like to suggest “economic inaccessibility” instead of “economic accessibility”

In Discussion section: It would be better to explore the possible reasons behind the fact that over two third of the participants were female, as well as operation hours of the PHC service centers. Discussing the implication of these findings and comparisons with other studies could be beneficial. For instance, consider addressing how these differences or similarities affect our understanding of hypertension prevalence, management and PHC utilization in your study population. Furthermore, discussing potential strategies, policy recommendation, and areas for further research to improve the situation would be valuable.

Line 296-298: Could you please provide more details on how the findings from this study support the idea that quality assurance of PHC services improves the utilization of these services for hypertension.

Reviewer #3: Congratulations for your efforts on this study with large sample size and strong methodology. Findings provide invaluable data for program implementation of PHC services. For more clarification on some points, the following comments are provided.

1.The affiliation of the corresponding author should be consistent. Is it Nay Pyi Taw or Pyin Oo Lwin?

2.In Figure (2), “twds” in the third predictor variable should be written in full words.

3.Abbreviation is needed for LHV in line 220.

4.It will be better if you can add the references for "the second reason for lower utilization in urban". For lines 245 and 246.

5.Please provide clarification on whether OOP is needed for both private and public facilities in Myanmar. Because the author’s discussion means to compare two facilities with the same purchasing mechanism for patients' health care. Do patients have to use OOP for all services from public health facilities? Line 249 “When spending OOP payments, people would choose private over public health 250 facilities”

6. PLOS authors have the option to publish the peer review history of their article (what does this mean?). If published, this will include your full peer review and any attached files.

Reviewer #1: No

Reviewer #2: No

Reviewer #3: No

---

## [Author Response · Author response to Decision Letter 0]

19 Sep 2024

Editor Comments:

We thank the editor for the useful and relevant comments and we respond as below: -

- The introduction needs to be restructured and improved. Furthermore, connections between paragraphs are not smooth. Information regarding the important role of primary care services in controlling hypertension is missing.

Yes, we have restructured the introduction to be smooth as you suggested. So, we have decluttered some sentences and added more information to be complete and concise. We have expressed the role of primary health care (PHC) in controlling hypertension and included sentences about feasibility and effectiveness of hypertension care in PHC, with the added reference.

- You mentioned 1001 participants, but from how many households? Was there any household replacement?

Yes, we made household replacement. Total 54 households got replaced for vacant households. We have added the process in methodology session of the manuscript. Of all total households, we got response from 1183 participants of 810 households. Our non-response rate was calculated based on number of participants. So, response rate was 85% (1001 out of 1183). It has been mentioned in revised manuscript.

- Can you differentiate whether the utilization is at facilities or through outreach according to the data? This information is important for public health interventions. For e.g. many of the participants utilized through outreach, then outreach interventions should be scaled up.

Unfortunately, no. The information where participants utilized the primary health services for hypertension could not be reported separately.

 But, according to findings, increased perceived travel cost to the nearest public health facilities decreased the utilization of PHC services provided by those facilities. Bivariate analysis also indicated that presence of public health facilities in their wards or villages increased the utilization of PHC services for hypertension. 

We have made conclusion from those two points that patients utilized PHC services if the PHC facilities were closer to their home or did not cost them to go the facilities whether it was mobile clinics or through outreach.

- Did you cover general practitioners (GPs)? Since GPs are important primary care providers in the community, if not, please consider changing the focus to 'utilization of public primary care services' and acknowledge the limitation of not covering GPs.

No, this study did not cover private PHC facilities including GPs. So, we have discussed it in revision with the added reference.

Reviewer #1: 

We thank the reviewer for giving comments and pointing out some points which need clarification. We respond as below: -

1. The abstract's methods section briefly describes the study design but lacks important details like how participants were selected and the response rate. 

We have revised the methodology section of the abstract with brief description of sampling method, response rate and the classification of blood pressure for hypertension.

2. The rationale for determining the sample size is well-explained, considering factors like confidence interval and margin of error. However, there's a discrepancy between the reported prevalence rates (88% vs. 28.9% from national estimates). To ensure accuracy, it's important to clarify how these differences were accounted for in the sample size calculation. 

We have made those two prevalence rates clear in revision that the prevalence of hypertension among population was 28.9% and the prevalence of hypertensive individuals who utilized primary health care facilities was 88%. 

3. The sampling procedure should address any potential biases introduced during participant selection.

We have discussed potential biases in the revised manuscript.

4. The definition of primary health care service utilization for hypertension is clearly defined, covering different types of health facilities and services. To strengthen this section, it would be beneficial to provide more details or validation methods on how utilization was measured in the health system.

We have mentioned details about validation methods for outcome measures as you suggested. 

5. It would be helpful to briefly mention any steps taken to protect participant confidentiality and ensure data security during and after data collection. 

We have added information about what we did for maintaining participant confidentiality in methodology section.

6. It would be good to clarify whether the study used age-standardization when comparing hypertension prevalence rates with other studies. 

We have compared this study prevalence with other studies prevalence rates after age-standardization. We have put this information in the revised manuscript. 

7. It would be better to expand on how these findings could inform the improvements of broader health policy and primary healthcare system in Myanmar. 

We have discussed some policy implications that would assist in strengthening of primary health system in discussion section.

8. Comparing findings with studies from other countries adds depth to the discussion. However, it's important to clearly explain how these comparisons were made, considering differences in healthcare systems and socio-economic factors. 

In the revised manuscript, we have made prevalence comparison with other Low-and-Middle income countries from South-East Asia with similar socioeconomic backgrounds. So, we have removed some prevalence rates of the countries which was mentioned in the previous manuscript, because of different socio-economic and geographical backgrounds (China, India and Nigeria) in making comparison.

Reviewer #2: 

We thank you the reviewer for insightful comments. We responded as below: -

In Financial Disclosure, it is mentioned that “The author(s) received no specific funding for this work”. However, the authors acknowledged the IR grant committee of Department of Medical Research, Ministry of Health. Could you please clarify whether this study was funded by the grant from IR grant committee of Department of Medical Research, Ministry of Health? If not, could you explain how the expenses for this study were covered.

We have got the IR grant from IR grant committee of Department of Medical Research, Ministry of Health. It covered for the expenses of data collection, but did not cover for the work of publication. 

I would like to suggest a through proofread. Some sentences are lengthy and could be rephrased for better clarity and readability.

Yes. We have modified some lengthy sentences into readable ones in the revised manuscript.

The abstract should serve as a standalone summary, including all essential details for comprehension without referring to the full manuscript. In this abstract, methods need to be elaborated further, and results and conclusion should be aligned. 

We have made sure this revised abstract is concise, complete and comprehending. We also have revised the methodology section to be complete, and results and conclusion sections to be aligned. 

We also have comments from another reviewer for the point of results not being aligned with conclusion.

We have modified it and discussed it in discussion section of the revised manuscript.

Line 29: Please specify the year in which stroke was the first leading cause of death in Myanmar 

It was the result of Verbal autopsy survey 2014-2016.

In Introduction section, it would be better to explain the reasons for choosing Pyin Oo Lwin Township and the population aged 40 years and above. Additionally, please clarify whether PHC services users need to pay for consultation fees or medicine.

According to PEN guidelines, 40 years and above is the target age group and it is considered as risk in calculating cardiovascular risk scores.

After PEN projects had piloted two townships of Yangon in 2012, PEN was expanded in 20 townships of 5 States and Regions in 2017. Pyin Oo Lwin was included in not only the extended townships, but also, piloted townships for Mandalay Region. 

In Myanmar Health Statistics 2019, it mentioned Mandalay was one of the Regions with higher prevalences of hypertension. Besides, whether hypertension services (including screening and treatment) provided at public health facilities of Pyin Oo Lwin Township were underutilized or not, was uncertain. This may be due to increased screening activities of the Region. 

However, number of people who utilize the services was much lower than that of 40 years and above township population according to Pyin Oo Lwin Township Public Health Department. That’s why we chose the study population was 40 years and above population of Pyin Oo Lwin Township.

Besides, PHC services provided at public health facilities do not need to pay for consultation fees and essential medicine if they are not stock-out. However, whenever there is shortage of medicines supply in health facilities, patients have to purchase medicines and medical supplies at retail pharmacies. costs patients with out-of-pocket expenditures. 

Line 51-59: it is better to provide descriptions of other PHC services providers in study site such as GP clinics, or outpatient departments which are not under the township public health department and should explain the reasons why these are not included as PHC services providers in this study.

This study did not cover private PHC facilities including GPs. So, we have discussed it in both methodology and discussion sections of the revised manuscript.

Line 71-79: It would be helpful to provide description of the location of the 7 villages and 3 wards.

We have added some information regarding the description of location of the study site in methodology section.

Line 42: Please correct the grammar for “PHC-oriented research were needed” 

Line 98: “Data was collected” should be “Data were collected”

Line 104: “two-parts” should be “two parts” 

Line 161: Please clarify “not aware of their statuses”. I assume “not aware of their hypertension”. 

Line 168: “Table (1) showed” should be “Table (1) shows” 

Line 181: “Shown in figure (2)” should be “As shown in figure (2)”.

Line 195: “Age composition” should be “The age composition” checked

Line 199: “30-79Years” should be “30-79 years” checked

Line 270: I would like to suggest “economic inaccessibility” instead of “economic accessibility” checked

Those mistakes have been corrected.

Line 173: Could you please clarify the rationale for using 150,000 MMK as the cutoff point? 

Until October 2023, Daily wages of a person was 4800 MMK. At least for single income households, they should be earned 150,000 MMK per month. It was similar to the income levels of Household Amenities in Myanmar (2014 - 2019), nearly over one third of households lived with annual income of 1.8 million MMK which was 150,000 MMK.

Line 185-186: Please clarify high perception and low perception. 

For better clarification, we have changed the terms “high” and “low” into “good” and “poor” perception. 

In this study, perception on hypertension is meant for a person’s own view or interpretation of control of hypertension. Perception on health facilities is meant for a person’s own view or interpretation that he or she has about public PHC facilities; how they stand in the community. Both perceptions are classified as poor (<50% scores), average (50-75% scores) and good (>75% scores). 

In Results section, it would be better to include relevant findings from bivariate analysis. Additionally, important negative findings-factors not associated with utilization of PHC services-should also be described and discussed in the main manuscript. 

We have discussed associations of outcome measures with some relevant findings in descriptive and bivariate analyses though they are not significant predictors in multivariate analysis. 

Line 207: Can you please spell out “SEAR” 

SEAR stands for South-East Asia Region. We have described it in abbreviations. We have also clarified it in the main manuscript.

Line 220: Please spell out LHV 

LHV stands for Lady Health Visitor. It has also been described in abbreviations. We have also clarified it in the main manuscript.

Line 226: I would like to suggest using “However” instead of “But” at the beginning of the sentence.

We have modified it according to your suggestion.

Line 248-249: Could you please elaborate more on “Consequently, urban residents choose health facilities based on their conveniences.”

We have modified that sentence in the revised manuscript. 

Secondary and tertiary public hospitals are mainly located in urban. Private health facilities (Private hospitals, Specialist clinics and GP clinics) are more condensed in urban. At the same time, urban had lower number of public primary health facilities than rural. Public hospitals are crowded and ambulatory OPD are not convenient with their working time. In addition, Myanmar has not established the mandatory tiered referral system.

At that time, urban residents choose health facilities based on their conveniences; which have geographical accessibility, affordability, sensitivity to their needs that meet to their satisfaction.

In Discussion section: It would be better to explore the possible reasons behind the fact that over two third of the participants were female, as well as operation hours of the PHC service centers. Discussing the implication of these findings and comparisons with other studies could be beneficial. For instance, consider addressing how these differences or similarities affect our understanding of hypertension prevalence, management and PHC utilization in your study population. Furthermore, discussing potential strategies, policy recommendation, and areas for further research to improve the situation would be valuable. 

We have ensured that, as a community-based cross-sectional study conducting in day time, male population are less likely to stay at home and participate in the study, unless we made the selection bias controlled; it was a limitation of the study. We mentioned other coinciding studies with references in the discussion section.

We thought the overall prevalence may have biased in any directions but, in Myanmar, male have higher behavioral risk factors than female. So, we inferred that low participation of male population may bring about underestimation of the overall prevalence.

We have recommended to prioritize NCDs care for economically active population at primary health care level for early detection and timely treatment. 

Line 296-298: Could you please provide more details on how the findings from this study support the idea that quality assurance of PHC services improves the utilization of these services for hypertension.

Thank you for pointing that out. We have this comment from another reviewer as well. 

Our study assessed the utilization of PHC services for hypertension. We found the utilization rate was low and we have discussed the possible reasons for low utilization and included about health seeking at higher quality of care and discussed it with references.

Moreover, we have also found that participants with good perception on hypertension and higher education were less likely to utilize the public health facilities. 

Kujawski et al. explored reasons for low utilization of public health facilities. In that study, households with hypertension tended to utilize private healthcare at facilities staffed with qualified providers.

That study also found that choosing private over public primary facilities was mainly because of technical quality reasons (poor quality of care, doctor not available, drugs not available, health personnel often absent, no adequate infrastructure). 

Opare-Addo et al. also found that public hospitals were preferred choices of health facilities because of their perception about offering good quality of care in higher level health facilities. That study also found that negative association of education with utilization of public health centers which agrees with our study findings.

After considering all the points, we have inferred that quality assurance of public PHC facilities; by continuous monitoring, assessment and improvement of essential health care infrastructure; improves the utilization of these services for hypertension

---

## [Decision Letter · Decision Letter 1]

2 Oct 2024

Prevalence of hypertension and factors associated with the utilization of primary health care services for hypertension among hypertensive population aged 40 years and above in Pyin Oo Lwin Township, Myanmar

PONE-D-24-18219R1

Dear Dr. Soe,

We’re pleased to inform you that your manuscript has been judged scientifically suitable for publication and will be formally accepted for publication once it meets all outstanding technical requirements.

Kind regards,

Kyaw Lwin Show, MPH, PhD

Academic Editor

PLOS ONE

Additional Editor Comments (optional):

Reviewers' comments:

Reviewer's Responses to Questions

**Comments to the Author**

1. If the authors have adequately addressed your comments raised in a previous round of review and you feel that this manuscript is now acceptable for publication, you may indicate that here to bypass the “Comments to the Author” section, enter your conflict of interest statement in the “Confidential to Editor” section, and submit your "Accept" recommendation.

Reviewer #1: All comments have been addressed

Reviewer #2: All comments have been addressed

Reviewer #3: All comments have been addressed

2. Is the manuscript technically sound, and do the data support the conclusions?

Reviewer #1: Yes

Reviewer #2: Yes

Reviewer #3: Yes

3. Has the statistical analysis been performed appropriately and rigorously? 

Reviewer #1: Yes

Reviewer #2: Yes

Reviewer #3: Yes

4. Have the authors made all data underlying the findings in their manuscript fully available?

Reviewer #1: Yes

Reviewer #2: Yes

Reviewer #3: Yes

5. Is the manuscript presented in an intelligible fashion and written in standard English?

Reviewer #1: Yes

Reviewer #2: Yes

Reviewer #3: Yes

6. Review Comments to the Author

Reviewer #1: (No Response)

Reviewer #2: Thank you for revising the manuscript and addressing most of the comments and recommendations provided in the first review.

Reviewer #3: The manuscript follows the guidelines and authors response thoroughly to the reviewers' comments and questions. Hope to be published soon.

7. PLOS authors have the option to publish the peer review history of their article (what does this mean?). If published, this will include your full peer review and any attached files.

Reviewer #1: No

Reviewer #2: No

Reviewer #3: **Yes: **MAY SOE AUNG, PhD (Public Health), MPH, MBBS, DipMedEd, Associate Professor, University of Medicine (1), Yangon, Myanmar

---

## [Editor Report · Acceptance letter]

7 Oct 2024

PONE-D-24-18219R1 

PLOS ONE

Dear Dr. Soe, 

I'm pleased to inform you that your manuscript has been deemed suitable for publication in PLOS ONE. Congratulations! Your manuscript is now being handed over to our production team.

Kind regards, 

on behalf of

Dr. Kyaw Lwin Show 

Academic Editor

PLOS ONE